# Counterfactual reasoning: Do Language Models need world knowledge for causal inference?

**Jiaxuan Li**
University of California Irvine
Irvine, CA 92617
`jiaxuan.li@uci.edu`

**Lang Yu**
Meta
Seattle, WA 98109
`langyu@fb.com`

**Allyson Ettinger**
University of Chicago
Chicago, IL 60637
`aettinger@uchicago.edu`

## Abstract

Current pre-trained language models have enabled remarkable improvements in downstream tasks, but it remains difficult to distinguish effects of statistical correlation from more systematic logical reasoning grounded on understanding of the real world. In this paper we tease these factors apart by leveraging *counterfactual conditionals*, which force language models to predict unusual consequences based on hypothetical propositions. We introduce a set of tests drawn from psycholinguistic experiments, as well as larger-scale controlled datasets, to probe counterfactual predictions from a variety of popular pre-trained language models. We find that models are consistently able to override real-world knowledge in counterfactual scenarios, and that this effect is more robust in case of stronger baseline world knowledge—however, we also find that for most models this effect appears largely to be driven by simple lexical cues. When we mitigate effects of both world knowledge and lexical cues to test knowledge of linguistic nuances of counterfactuals, we find that only GPT-3 shows sensitivity to these nuances, though this sensitivity is also non-trivially impacted by lexical associative factors.

## 1 Introduction

Reasoning plays a central role in human communication [5]. While language models have demonstrated remarkable ability on downstream tasks [2, 11, 6], it remains unclear to what extent predictions generated by language models are consequences of correlation with linguistic heuristics in the context, versus robust reasoning about causal relations grounded on understanding of world knowledge.

In this paper we leverage *counterfactual conditionals* to investigate the capacity of pre-trained LMs (PLMs) to distinguish hypothetical scenarios from reality, and to examine how this interacts with models' use of existing real world knowledge as well as shallower associative cues. Counterfactuals consist of a premise which is false in real world but true in the hypothetical world (e.g., *If cats were vegetarians*), and an imaginary consequence of this premise (*cats would love carrots*). Testing language models with counterfactuals allows us to use language to manipulate what is true and what is hypothetical, and to test models' ability to separate and use this information for predictions. Previous work has established the use of counterfactual scenarios to probe inference ability [10, 14, 8, 7, 12], but the datasets lack systematic control of lexical cues and world knowledge, which makes it likely that the performance could be attributable to spurious cues in the datasets [9].

For our tests we draw on and adapt inputs from existing psycholinguistic experiments. We begin by testing models' ability to override existing world knowledge when the context indicates that the correct completion involves a hypothetical world (e.g., "if cats were vegetarian, cats would eat *carrots/fish*"). We test five popular PLMs, and find that models can increase their preference for counterfactual completions given counterfactual contexts—however, these increased preferences remain around chance, and further inspection indicates that most models rely strongly on simple

36th Conference on Neural Information Processing Systems (NeurIPS 2022).

lexical cues. Next we remove the influence of real world knowledge and mitigate effects of lexical triggers, to test models' understanding of what counterfactual language implies about the world state. We find that most models fail to understand real-world implication of counterfactuals and largely rely on lexical triggers—with the exception of GPT-3, which shows greater sophistication, but continues to show non-trivial susceptibility to interference from lexical-associative cues. We discuss the implications and possible interpretations of these findings with respect to linguistic sophistication and strategies of these models.[1]

## 2 Testing models on overriding world knowledge

Our first experiment investigates whether LMs are able to take a counterfactual scenario (e.g., "if cats were vegetarian"), and predict a subsequent completion that is consistent with the counterfactual scenario but that contradicts general world knowledge (e.g., "families would feed cats with carrots").

**Items**  For this experiment we take direct inspiration from the psycholinguistic study of [4]. There are 128 items from the original psycholinguistic experiments, and we synthetically generated 10,720 additional items. The design of the synthetic dataset is similar to the psycholinguistic stimuli, but we use a variety of syntactic constructions to describe a given event, as well as varying lexical selections, tense markers and modal verbs (see Appendix A.1 for illustration of data generation process). We matched the target nouns and syntactic constructions across conditions so that we could control lexical properties that influence language models' predictions.

The experiment includes two key conditions: Counterfactual-World (CW) and Real-World (RW). Examples are shown in Table 1.

Table 1: Example items in CW, RW, BB conditions.

| Condition | Sentence |
| --- | --- |
| CW | If cats had liked vegetables, they would be cheaper to keep. Families would feed their cats with ***carrots/fish***. |
| RW | Because cats like meat, they are expensive to keep. Families would feed their cats with *carrots/**fish***. |
| BB | Families would feed their cats with *carrots/**fish*** |

The example in the CW condition presents a counterfactual scenario in which cats are vegetarians—as a result the logical target completion is "carrots", but because cats are in reality more likely to eat fish, this contradicts world knowledge. The RW condition complements the CW condition, providing a baseline in which the logical completion ("fish"') is consistent with the real world.

We also include one additional, simpler Baseline Bias condition (BB), for a more direct test of the strength of models' baseline preference for factual versus counterfactual completions, in the absence of context guiding which is relevant. An example is shown in Table 1.

**Experiments**  We investigate the counterfactual reasoning ability in five pre-trained language models. We include autoregressive transformers in the GPT family (GPT-2 [11] and GPT-3 [1]) and masked language models in the BERT family (BERT [2], RoBERTa [6] and MPNet [13]).

We test models by comparing the log-probability that each model assigns to the counterfactual ("carrots") and factual ("fish") completions given the contexts. For all conditions, we compute the percentage of items in which the counterfactual continuation has a higher probability than the factual continuation. Note that by contrast to CW, in RW and BB conditions the counterfactual completion is the less logical completion, so *lower* values in these two conditions reflect better predictions.

**Results**  Table 2 shows the preferences for counterfactual completions across all models and conditions, for the small-scale hand-designed items from the psycholinguistic experiment, and the

---

[1]We make all data and code available at (https://github.com/goldengua/Counterfactual_Inference_LM) for future testing.

Table 2: Preference for counterfactual completion (e.g., *carrots*) in CW, RW, BB conditions.

| Model | Small-scale | | | Large-scale | | |
|---|---|---|---|---|---|---|
| | CW | RW | BB | CW | RW | BB |
| GPT2 | 53.1 | 34.4 | 40.6 | 53.7 | 29.5 | 31.5 |
| GPT3 | **68.8** | **18.8** | **18.7** | **71.3** | **2.5** | **14.7** |
| BERT | 46.9 | 43.8 | 31.2 | 34.2 | 14.3 | 35.2 |
| RoBERTa | 53.1 | 21.9 | 21.9 | 61.4 | 26.5 | 47.2 |
| MPNet | 50.0 | 21.9 | 21.9 | 66.9 | 15.6 | 36.6 |

large-scale synthetic items. We see that all models show stronger preference for counterfactual continuations in the counterfactual (CW) context (though in the case of BERT on the small-scale data, this difference is negligible). All models show below-chance preference for counterfactual continuations in the RW conditions—which means above-chance preference for the correct factual continuations. However, though all model preferences for the correct counterfactual continuation are higher in the CW condition than in the RW condition, even in the CW condition the preference for counterfactual conditions is at best slightly above chance for most models. The exception is GPT-3, which is the only model to prefer the counterfactual continuation in greater than 70% of items.

We also see that models that have stronger factual preferences—that is, lower counterfactual preferences—in the Baseline Bias condition (GPT-3, RoBERTa, MPNet) also show stronger increase in preference for the counterfactual in the CW condition. This suggests, slightly counter-intuitively, that stronger grasp on relevant world knowledge may in fact be associated with models *more* effectively overriding that knowledge in a counterfactual. To investigate this effect further, we examine the impact of world knowledge at the item level. We quantify strength of world knowledge for a given item based on difference between models' log-probability of counterfactual and factual continuations for that item in the BB condition. We then compute the Pearson correlation between these differences and correctness in the CW condition. We find a significant correlation between the robustness of world knowledge encoding and correctness of prediction (see Appendix A.2 for details), further supporting a relationship between strength of world knowledge and sensitivity to the counterfactual.

## 3 Testing impact of cue words in context

The results above suggest that models can to an extent override world knowledge in the presence of a counterfactual, particularly in cases when models have a strong handle on the relevant world knowledge. However, it is possible that in these tests the models were not relying on sophisticated understanding of counterfactuals, but rather on simple lexical triggers in context. Consider, for instance, that models could perform well above if they simply increase their preference for "carrots" in the proximity of "vegetables" and for "fish" in the proximity of "meat". To test the impact of these lexical triggers, we incorporate an additional condition described below.

**Items** Table 3 shows a sample item. In this Counterfactual-to-Reality (CR) condition, models see the same counterfactual context, but the subsequent sentence references actual reality. So the correct completion is consistent with reality, but inconsistent with the lexical trigger ("vegetables").

Table 3: Example items in CR condition.

| Condition | Sentence |
|---|---|
| CR | If cats had liked vegetables, they would be cheap to keep. In reality, families feed their cats with ***fish/carrots***. |

**Experiments** As above, we calculate percentage of items in which models prefer the counterfactual over factual continuations. Models that are relying on linguistic information beyond simple lexical triggers should show a sharp drop in preference for the counterfactual completion in the CR condition, where the correct completion should align with real world information.

**Results** Table 4 shows the results. We see that most models show non-zero drop between CW and CR conditions—however, for most models this reduction is minor. It is only GPT-3 that shows a truly substantial drop in counterfactual preference, and only in the large-scale synthetic dataset. This suggests that most models are largely following the lexical triggers, while GPT-3 has somewhat greater sensitivity to more detailed linguistic cues. Note, however that GPT-3's relative success on the synthetic data over the small-scale data may rely on larger distance between the lexical trigger and the target position: see Appendix A.3 for example items and evidence of GPT-3's sensitivity to distance between lexical trigger and target word.

Table 4: Percentage of counterfactual completion in CW and CR condition.

| Model | Small-scale | | Large-scale | |
|---|---|---|---|---|
| | CW | CR | CW | CR |
| GPT2 | 53.1 | 50.0 | 53.7 | 51.9 |
| GPT3 | **68.8** | 56.2 | **71.3** | **28.0** |
| BERT | 46.9 | 46.9 | 34.2 | 39.4 |
| RoBERTa | 53.1 | **37.5** | 61.4 | 57.3 |
| MPNet | 50.0 | 46.9 | 66.9 | 58.1 |

## 4 Testing models' use of counterfactual cues to infer world state

The previous experiments indicate that models are able to override world knowledge in the face of counterfactual evidence, and that the ability to do this improves with stronger world knowledge—but that for most models this performance is driven largely by lexical triggers in the context, with the possible exception of GPT-3. In this section we remove the influence of pre-existing world knowledge, and hold constant lexical triggers across conditions, for a more direct test of models' sensitivity to linguistic indicators of counterfactuals, and what they say about the true state of the world. This task is particularly challenging as the prediction requires language models to infer the true state of the world based on counterfactuals, in which case the lexical cues are often misleading.

**Items** We adapt stimuli from a psycholinguistic study in which frequency of target completions is controlled [3]. There are 96 sentences in the dataset. We additionally create a larger-scale synthetic dataset with 6,480 sentences in each critical condition. The dataset uses the same events as the generated dataset from Section 2, but we modify the subject noun phrase such that there is no influence of existing world knowledge. We do this via examples in which existing world knowledge cannot inform the correct completion—instead, models simply need to infer based on the counterfactual language that the true state of the world is different from what the counterfactual states. Further, we fully control the lexical items used across different conditions to minimize the lexical effect. For this purpose, we use sentences like those in Table 5.

In the Counterfactual-World Context (CWC) condition, the scenario described in the first sentence is neutral with respect to real world knowledge—it is the use of the counterfactual that tips us off that that this scenario is not true in reality. The correct completion here cannot be informed by world knowledge, and is also misaligned with the lexical trigger (e.g., "vegetables"), so models must rely specifically on this implication of the counterfactual in order to perform well.

In the Real-World Context Alternative (RWCA) condition, the context uses the same lexical triggers as the CWC condition. However, the logical completion is different from CWC condition, since the word ("carrots") associated with the lexical trigger ("vegetables") is the logical completion.

Given that the logical completions in CWC and RWCA are different, we also compare this against a Baseline Bias Context (BBC) condition, in order to establish default model preference for the target factual completion in the presence of the new subject noun phrase.

**Experiments** We compare proportion of CWC-congruent completions across conditions. Good models should assign high values in the CWC condition and low values in the RWCA condition.

Table 5: Example items in CWC, RWCA and BBC condition.

| Condition | Sentence |
|---|---|
| CWC | If the pet had loved vegetables, it would be very surprising. In fact, people feed the pet with ***fish**/carrots*. |
| RWCA | Because the pet loved vegetables, it was very surprising. In fact, people feed the pet with *fish**/carrots***. |
| BBC | In fact, people feed the pet with *fish/carrots*. |

Table 6: Percentage of CWC-consistent completion ("fish") in CW, RWCA and BBC condition.

| Model | Small-scale | | | Large-scale | | |
|---|---|---|---|---|---|---|
| | CWC | RWCA | BBC | CWC | RWCA | BBC |
| GPT2 | 66.7 | 66.7 | 33.3 | 35.8 | 32.2 | 72.6 |
| GPT3 | **62.5** | **33.3** | 50.0 | **47.6** | **32.2** | 73.8 |
| BERT | 45.8 | 33.3 | 50.0 | 53.0 | 53.0 | 71.5 |
| RoBERTa | 50.0 | 50.0 | 50.0 | 35.7 | 31.3 | 72.5 |
| MPNet | 37.5 | 33.3 | 62.5 | 41.4 | 32.3 | 68.5 |

**Results**   Table 6 shows the results. In the small-scale dataset, most models show a similar preference in CWC and RWCA, suggesting again that their predictions are largely driven by lexical triggers. Only GPT-3 shows substantial difference between CWC and RWCA, indicating finer-grained sensitivity to counterfactual structures. This sensitivity is, however, less pronounced in the large-scale dataset. Closer inspection suggests that GPT-3's particular success on the small-scale data may in fact be attributable to canceling out of lexical triggers. For example, in the sentence "*If Helen had received her first student loan, her bank balance would now be in credit. When she checked her bank balance today she was **worried/happy** with her financial situation.*", there are lexical triggers supporting both continuations, which may cause lexical factors to cancel out, allowing more influence from other linguistic cues. By contrast, in the large-scale dataset, the lexical trigger ("vegetables") always favors the CWC-inconsistent continuation ("carrots"), causing strong lexical bias against the CWC-congruent continuation (see Appendix A.3 for further analysis on the role of conflicting lexical triggers and other linguistic factors). This suggests that GPT-3 does show real sensitivity to linguistic indicators of counterfactuals, but the influence of superficial associative cues remains strong.

## 5   Conclusion

The experiments above have shown that when presented with counterfactual situations, PLMs are consistently able to prefer completions that conflict with world knowledge—and counterintuitively, this sensitivity appears better in cases where that world knowledge is stronger. Our results also indicate, however, that models are in large part relying on simple lexical cues to inform these preferences. The only model that shows more sophisticated sensitivity to fine-grained linguistic cues separating counterfactuals from reality is GPT-3—which successfully distinguishes conditions based on counterfactual cues, but nonetheless still shows strong influences from lexical associative cues. So how do we interpret these findings? Why does world knowledge aid counterfactual sensitivity? Does GPT-3 truly understand counterfactuals? One possibility worth considering is that explanations in both of these cases involve volume of exposure. First, models' stronger world knowledge for a given fact suggests that models have encountered that fact more often in training—and this may in turn translate to more exposure to that type of knowledge in counterfactual contexts, enabling more straightforward memorization-based performance. Similarly, while GPT-3 may robustly understand counterfactuals, the massive data exposure for that model may enable a simpler path to success in these experiments: GPT-3 may simply have developed lower-level knowledge of how linguistic cues like "If/had" versus "Because" mediate how closely lexical triggers in context will associate with later words. We leave investigation of these hypotheses for future work.

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

# A   Appendix

## A.1   Generation process of dataset

Table 7 shows the illustrative process of data generation. We first manually design a template of an event (e.g. love-feed: $subject_1$ love $object_1$, $subject_2$ feed them with $object_2$). Then we vary the selection of subjects and objects and combine them in various ways. Tense and modal of the verbs are also modulated.

Table 7: Illustrative representation of data generation process in large-scale synthetic dataset. Different sentences could be generated based on the *original* sentence by changing the lexical item of subjects and objects.

| Condition | Sentence |
|---|---|
| Original | If cats had loved vegetables, families would feed them with carrots. |
| Subject$_1$ | If *dogs* had loved vegetables, families would feed them with carrots. |
| Object$_1$ | If cats had loved *greens*, families would feed them with carrots. |
| Subject$_2$ | If cats had loved vegetables,*breeders* would feed them with carrots. |
| Object$_2$ | If cats had loved vegetables, breeders would feed them with *cabbages*. |
| Modal | If cats had loved vegetables, families *might* feed them with carrots. |
| Tense | If cats *loved* vegetables, families would feed them with carrots. |

## A.2 Correlation with world knowledge

Table 8 shows the correlation between the robustness of world knowledge representation and the correctness of counterfactual predictions. Across all language models, there is a significant correlation with correlation coefficient greater than 0.47, indicating that language models benefit from a good representation of world knowledge.

Table 8: Correlation between robustness of world knowledge encoding and correctness of counterfactual predictions. $p < .05$*, $p < .01$**, $p < .001$***.

| Model | Small-scale | | Large-scale | |
|---|---|---|---|---|
| | *coef* | *p* | *coef* | *p* |
| GPT2 | .75 | <.001*** | .56 | <.001*** |
| GPT3 | .49 | .004** | .60 | <.001*** |
| BERT | .55 | .001** | .47 | <.001*** |
| RoBERTa | .47 | .006** | .47 | <.001*** |
| MPNet | .63 | <.001*** | .49 | <.001*** |

## A.3 Testing impact of linguistic cues on GPT-3

What kind of linguistic factors is driving the performance of GPT-3? We further designed a series of sentences by varying different features and tested them with GPT-3. Each condition in the dataset has 100 sentences. The first test is a subset of the synthetic large-scale dataset featuring CWC, RWCA, BBC conditions listed above. Similar to the previous result, GPT-3 shows some extend of reasoning by preferring CWC-consistent continuations, even when context words are very similar (see Table 9).

Table 9: Percentage of CWC-consistent continuation ("fish") in CW, RWCA and BBC condition in synthetic subset.

| Condition | Sentence | GPT-3 |
|---|---|---|
| BBC | In fact, people feed Mary with *fish/carrots*. | 42.5 |
| CWC | If Mary had loved vegetables, it would be very surprising. In fact, people feed Mary with ***fish**/carrots*. | 34.8 |
| RWCA | Because Mary loved vegetables, it was very surprising. In fact, people feed Mary with *fish/**carrots***. | 27.3 |

We further test to what extend GPT-3 could cancel out the effect of lexical cues by inserting conflicting lexical cues in the context using the discourse connective "rather than". Though a conflicting lexical cue appears, the context-consistent completion should remain the same. Table 10 shows that GPT-3 is greatly affected by the presence of conflicting lexical cues. After inserting "*meat*" into context, the percentage of CWC-consistent continuation ("*fish*"), indicating a strong lexical effect of the presence of a conflicting cue.

Table 10: Percentage of CWC-consistent continuation ("fish") in CW, RWCA and BBC condition in synthetic subset.

| Condition | Sentence | GPT-3 |
|---|---|---|
| CWC (Rather) | If Mary had loved vegetables rather than meat, it would be very surprising. In fact, people feed Mary with *__fish__/carrots__*. | 48.5 |
| RWCA (Rather) | Because Mary loved vegetables rather than meat, it was very surprising. In fact, people feed Mary with *fish/**carrots***. | 47.0 |

Next, we test to what extend to salience of lexical cues is related to its distance to the target word. We bring the conflicting lexical cues to the beginning of the sentence with the discourse connective "instead of". After moving the conflicting cue "meat" further from the target word, it is less likely to predict "fish" in both conditions (Table 11). The result suggests that linear distance from lexical cues to the prediction target has a strong impact.

Table 11: Percentage of CWC-consistent continuation ("fish") in CW, RWCA and BBC condition in synthetic subset.

| Condition | Sentence | GPT-3 |
|---|---|---|
| CWC (Instead) | If instead of meat, Mary had loved vegetables, it would be very surprising. In fact, people feed Mary with *__fish__/carrots*. | 28.5 |
| RWCA (Instead) | Because instead of meat, Mary loved vegetables, it was very surprising. In fact, people feed Mary with *fish/**carrots***. | 33.8 |

Finally, we probe how linguistic markers enhance the correct prediction of counterfactual sentences. We test the effect of sentence boundary (indicated by period), discourse connectives (indicated by "In fact") and tense. GPT-3 shows a fair sensitivity to linguistic markers. For linguistic markers that would shift the logical completion from "fish" to "carrots", GPT-3 is less likely to select "fish" as a preferred continuation, and tense is the most salient effect on the preference shifting. For discourse connective "in fact" that is going to increase the preference of "fish", GPT-3 shows a slightly larger preference for "fish".

Table 12: Percentage of "fish" in CW, RWCA and BBC condition in the presence of a conflict cue connected with "instead of".

| Condition | Sentence | GPT-3 |
|---|---|---|
| Period | If Mary had loved vegetables, it would be very surprising, in fact, people feed Mary with *fish/**carrots***. | 28.7 |
| Connectives | If Mary had loved vegetables, it would be very surprising. People feed Mary with fish/carrots *__fish__/carrots*. | 35.5 |
| Tense | If Mary had loved vegetables, it would be very surprising. In fact, people would feed Mary with *fish/**carrots***. | 14.0 |