# OpenReview forum: "Counterfactual reasoning: Do Language Models need world knowledge for causal inference?"
_NeurIPS.cc/2022/Workshop/nCSI — nCSI WS @ NeurIPS 2022 Poster_

### Official Review · Reviewer_5Po1 · 2022-10-12
**An interesting perspective on language models**

**Rating:** 2
**Confidence:** 3

**Review:**

**Summary**: This paper experiments on several state-of-the-art pre-trained language models to see if they can predict unexpected counterfactual consequences arising from hypothetical situations. This is accomplished by providing a text prompt that begins by establishing a hypothetical premise (inconsistent with reality), then comparing the model’s likelihood outputs of two possible continuations, one consistent with reality and the other consistent with the hypothetical world. The result is compared with baselines of either using reality as the premise or by having no contextual premise. Additional experiments are performed to evaluate the effects of cue words. Results show that, when accounting for cue words, GPT-3 is the only model that shows significant understanding of counterfactual situations, but results are largely random in all cases.

**Strengths**:
1. This paper presents an unusually novel idea which offers valuable insight into the evaluation of large language models. The unique experimental results are very revealing about the true nature of such models and provide much needed critical analysis.

2. The several examples added in the paper greatly contributed to the readability of the paper. Although the ideas of “counterfactual conditionals” are complicated and not easily defined, the examples provided strong intuition of what to expect in each experimental setting.

3. The experiments are very extensive and well thought-out.

**Weaknesses**:
1. The paper is quite subjective. The idea of a “counterfactual conditional” is not rigorously defined and relies on humans to decide whether a conditional in text is considered factual or counterfactual. As such, data points in the dataset are handcrafted by humans and are subjectively determined by humans to be appropriate for this task. Then the evaluation of the models may not have a clear meaning.

2. Even in the experiments in which the effects of lexical cues are considered, success of a model at differentiating between the two cases does not seem to necessarily imply any knowledge of counterfactual logic. Perhaps a powerful language model could simply understand lexical cues more precisely. For example, seeing the word “if” makes the condition more hypothetical (and should therefore be ignored), and seeing the phrase “in reality” implies a true continuation.

3. Generally, counterfactual information is impossible to obtain because one cannot observe two realities at once. Hence, there will be a lack of counterfactual data in the training data for these large language models. Then, there may be little confidence that the language models will be correct, even if they achieve perfect results on the experiments in the paper. For example, we can hypothesize that cats would eat carrots if they were vegetarian, but perhaps they would hate carrots too. In other cases, it may not even be clear that the result would be different in the hypothesized world compared to the real world.

In general, although the weaknesses listed above should be considered as heavy limitations of this work, I think the ideas presented in this paper offer interesting insight into language models and would be a great contribution to the workshop.

---

### Official Review · Reviewer_swis · 2022-10-15
**The paper adds an additional point of evidence towards understanding the reasoning capabilities of (L)LMs.**

**Rating:** 2
**Confidence:** 2

**Review:**

## Summary

In their paper the authors test the abilities of different language models with regard to their ability of integrating counterfactual information in their decision process. Section 2 contains experiments about integrating provided counterfactuals into the model decision and measuring for common or counterfactual text completion. Section 3 inspects whether the LM completions are based on 'real' text understanding or simple correlations to cue words of the same domain. Section 4 tests whether LMs can infer counterfactuals in scenarios without applicable world knowledge by providing the models with sentences stating 'surprising' observations - indicating that the opposite statement should be inferred as the expected answer. All experiments are conducted using a small hand-crafted dataset and a larger, synthetically extended, version.

## Related work

The paper seems to cover relevant related work with regard to querying information from LMs and inspecting counterfactuals.

## Strengths

* The authors present a set of straight forward experiments for querying LMs for counterfactual information using three different experimental setups in the main paper. The experiments take into account possible influences of word domain correlations as well as the influence (or absence) of previous world knowledge. The experiments are well designed to answer the posed questions.
* The results are well presented and reveal significant differences in text understanding, especially, between GPT-3 and the other models.

## Weaknesses

While synthetic dataset samples are constructed to match the style of the smaller dataset, we see strong deviations between the small- and large-scale datasets results. E.g. Table 6 BBC results change from the expected random guessing (50%) in the small-scale dataset to 70% in the large-scale one. In my opinion the paper could be improved by inspecting and explaining which parts of the synthetic samples are responsible for this shift in performance. Overall, the paper could be improved by further analyzing which factors of the presented datasets appear to be more challenging/difficult to the LMs.

### Additional Comments

Results for table 2 large-scale RW reports an exceptionally good performance for GPT-3 in comparison to the other models. I would like the authors to confirm whether the reported value is correct and may suggest discussing it in the paper.

## Conclusion

The paper provides an overview over the reasoning capabilities of different LM within the domain of inferring counterfactual knowledge from text. The authors provide three empirical evaluations in the main paper that inspect the incorporation of counterfactual information into the LM reasoning process while testing against erroneous influence of lexical cues. With the upcoming popularity of LLMs, this research contributes towards a better understanding of LM capabilities and adds to the pile of evidence that helps the field of identifying the strengths and weaknesses of LM text understanding and reasoning capabilities.

---

### Meta-Review · Area_Chair_5jJX · 2022-10-17

**Recommendation:** 2
**Confidence:** 3

**Metareview:**

Paper investigates whether LMs need world knowledge for causal inference. This is an interesting question. The experimental results  show that there is still quite some work to be done. For instance, GPT-3 distinguishes conditions based on counterfactual cues, but nonetheless still shows strong influences from lexical associative cues. Moreover, world knowledge seems to aid counterfactual sensitivity. However, I really like that the authors are critical, arguing that GPT-3 may simply have developed lower-level knowledge of how linguistic cues like “If/had” versus “Because” mediate how closely lexical triggers in context will associate with later words. It would be interesting to check in future work whether humans are also making use of similar triggers. Anyhow, the reviewers also agree that this paper should be accept. Overall and interesting contribution.

---

### Decision · Program_Chairs · 2022-10-20

Accept (Poster)